# Dense Multiscale Feature Learning Transformer Embedding Cross-Shaped Attention for Road Damage Detection

Chuan Xu [1,†], Qi Zhang [1], Liye Mei [1,†], Sen Shen [2], Zhaoyi Ye [1], Di Li [1], Wei Yang [3] and Xiangyang Zhou [3,*]

1   School of Computer Science, Hubei University of Technology, Wuhan 430068, China
2   School of Weapon Engineering, Naval Engineering University, Wuhan 430032, China
3   School of Information Science and Engineering, Wuchang Shouyi University, Wuhan 430064, China
*   Correspondence: zhouxiangyang@wsyu.edu.cn
†   These authors contributed equally to this work.

**Abstract:** Road damage detection is essential to the maintenance and management of roads. The morphological road damage contains a large number of multi-scale features, which means that existing road damage detection algorithms are unable to effectively distinguish and fuse multiple features. In this paper, we propose a dense multiscale feature learning Transformer embedding cross-shaped attention for road damage detection (DMTC) network, which can segment the damage information in road images and improve the effectiveness of road damage detection. Our DMTC makes three contributions. Firstly, we adopt a cross-shaped attention mechanism to expand the perceptual field of feature extraction, and its global attention effectively improves the feature description of the network. Secondly, we use the dense multi-scale feature learning module to integrate local information at different scales, so that we are able to overcome the difficulty of detecting multiscale targets. Finally, we utilize a multi-layer convolutional segmentation head to generalize the previous feature learning and get a final detection result. Experimental results show that our DMTC network could segment pavement pothole patterns more accurately and effectively than other methods, achieving an F1 score of 79.39% as well as an OA score of 99.83% on the cracks-and-potholes-in-road-images-dataset (CPRID).

**Keywords:** road damage detection; cross-shaped attention; dense multi-scale feature learning

## 1. Introduction

The infrastructure of the road is an important public resource that contributes to economic development and growth, while providing a number of social benefits. The aging of roads, the rapid increase in the number of vehicles, and frequent use result in damage to the road surface, which creates various defects in many countries' roads. Over time, pavement damage becomes a common phenomenon and it affects people's lives to varying degrees, with poor road conditions leading to excessive wear and tear on vehicles, as well as increasing the likelihood of collisions and delays that can lead to traffic accidents [1]. Road surface defects are of different shapes, sizes, numbers and varying degrees of damage, which is a challenge for researchers of road damage. Moreover, the interference of numerous environmental factors makes it difficult for researchers to detect and repair these defects in real time. As a result, it is important to effectively detect the road pavement condition and to develop an automated intelligent road damage detection algorithm, in order to achieve rapid detection. This is important for managing, protecting and repairing the poorer road surfaces, and is a pressing issue for improving road safety.

Recently, many researchers focus on the field of computer vision for road damage recognition, and there are two main types of methods: traditional methods and deep learning methods. Traditional methods include threshold segmentation, edge detection, mathematical morphological operations, etc., and the combination of these algorithms

with traditional machine learning classifiers such as artificial neural network (ANN) and support vector machine (SVM). However, it is difficult to accurately detect cracks in real pavement environments due to noise effects such as illumination variations and ground roughness. With the technical progress of general-purpose deep learning methods [2], many researchers apply deep learning-based detection methods to the task of detecting road damage, for instance image classification [3,4], target detection [5–7], and semantic segmentation [8–11]. These algorithms are effective in the detection of road damage [12–14]. Among them, the detection method based on image classification is to first segment the original image into sub-image fast, then judge these sub-image blocks by using a binary classification network; a final step involves stitching these sub-image modules into the original image. Nevertheless, this type of approach has the disadvantages of ignoring the relationship between the sub-image blocks and the surrounding environment, small acceptance field, and unsatisfactory detection results [15]. There is widespread use of detection methods based on target detection, such as Faster R-CNN network [16–18] and YOLO series networks [19,20], but this detection model requires layer-by-layer down-sampling, resulting in poor model recognition of fine targets [21–23]. The semantic segmentation-based method of pavement damage detection detects pavement damage by determining whether each image element in the image is a cracked image element, which provides an accurate assessment of pavement condition. We often use it for road damage detection tasks. For example, the CrackNet series network [24–26] addresses crack detection in 3D pavement data [27], FCN network achieves an end-to-end effect in crack detection [28–30], and a U-net series network [31,32] proposes a unique data enhancement approach and boundary weighted loss function for crack detection using encoding-decoding structure. The DeepLab family of networks [33–35] combines null convolution and multiscale information to enhance network performance for crack detection. The above review shows that existing CNN-based network achieves good experimental results, but such models mainly rely on convolution and down-sampling to obtain large perceptual field information, which can only obtain short-range correlations and thus are usually ineffective in modeling long-range dependencies. In spite of the addition of the dilation/void convolution or attention module, the main network architecture remains the same, and it is only possible to improve the model performance to a limited extent [36]. In contrast to CNN, the transformer model represents coarser spatial information through location encoding and tokens, and uses stacked transformer blocks to obtain features. It not only dynamically adjusts the sensory field, but also obtains more long-range global correlation information. The CrackFormer network uses a novel attention module to detect fine-grained cracks [37], UCCrack network introduces vision transformer-based cross-attention for automatic recognition of road cracks [38], and Swin-Transformer combines convolutional neural networks and vision transformers for identifying road damage accurately [39]. Comparing with CNN, the new Transformer technique has the advantages of strong ability to learn long-distance dependence and multimodal fusion, but it is also more difficult to extract the target features for different scale changes, and recognition detection is not effective.

We propose a brand-new deep learning-based model, dense multiscale feature learning Transformer embedding cross-shaped attention for road damage detection (DMTC), for detecting pothole patterns in road images. Our network uses an encoder-decoder architecture to identify features in road images using the cross-shaped attention (CSA) mechanism in the encoding stage. In this case, the cross-shaped attention block (CSAB) uses a CSA mechanism to achieve global attention [40]. In addition, we add a module for locally enhanced positional encoding to our self-attention branch, and add positional encoding to the self-attention operation by operating on ignored positional information in each block. The design decouples the position encoding from the self-attentive calculation, which allows increasing the local sensing bias. In the decoding stage, we use a dense multiscale feature learning (DMFL) module to ensure that no information is lost during the fusion of networks. The module achieves the fusion of multi-scale feature information through the structure from top down and lateral connection fusion, which effectively solves

the difficult problem of small potholes detection [41]. Contributions of this paper include the following:

(1) We use CSA mechanisms in the backbone network and focus on the pothole region to expand the attentional action range, which enables our DMTC network to deploy global attention to the specified feature information more efficiently, enhances the representational capabilities of the network, makes the network more capable of detecting environmental and road potholes, and thus improves the accuracy of the network's recognition.

(2) We utilize the DMFL module to fuse independent information of multiple scales, which significantly improves the detection performance. The DMFL module quickly constructs a feature pyramid that contains strong semantic information at every scale, recovers as much of the original pothole feature information as possible, reduces our DMTC model's false detection rate, and makes the edge lines of the detected potholes are more complete.

(3) On the publicly available road detection dataset CPRID, we replicate some segmentation algorithms, provide baselines for road damage, and conduct extensive experiments. Results of the experimental work in this paper demonstrate that our method is visually and quantitatively superior in comparison with other conventional methods.

## 2. Materials and Method

For the multi-scale feature information of potholes in road damage images, we propose a DMTC network model to segment road damage effectively. Figure 1 shows the three main components of our network: CSAB feature extraction module, DMFL module, and Segmentation-Head module. Among them, the CSAB feature extraction module uses the cross-attention module to create long-range interactions between the underived feature maps, which extends the attention range and thus improves the feature representation of the damage. The DMFL module uses the fusion of the shallow layer with high resolution and the deep layer with rich semantic information to fuse the feature maps at different stages to obtain a fine-grained result. The Segmentation-Head module uses multi-layer convolution to perform the final generalization and learning of the feature map and output the final result. The following sections will present the three modules in detail.

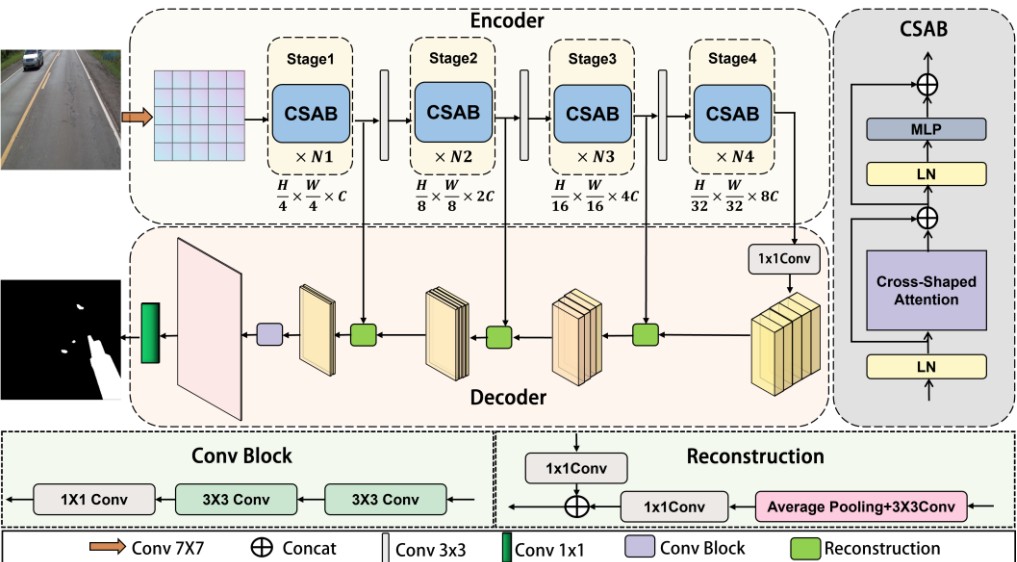

**Figure 1.** Overview of the proposed DMTC. Our network receives road images as input. The CSAB module uses a cross-attention mechanism to obtain bit-time multiscale feature information with rich contextual semantics. The DMFL module progressively integrates the extracted multiscale feature information. Finally, the road pothole segmentation results are output through a fully connected convolutional network.

### 2.1. Cross-Shaped Attention

Road damage detection is a challenging problem, which due to uneven image intensity, complex topology, low contrast and pothole background noise. In addition, the diversity of potholes enhances the difficulty of detection. The CSAB model collects contextual information both horizontally and vertically to enhance pixel-level representation and is more efficient than non-local blocks. It also has a sufficient number of heads of multi-headed self-attentive layers, which are as expressive as any convolutional layer. Therefore, we utilize the CSAB module to encode feature information from images of road potholes.

We input an original image $C_1$ and pass it through a convolutional layer with a convolution kernel size of $7 \times 7$ and a step size of 4, and at this point there are 96 channels, thus changing the number of input channels. Throughout the entire decoding module, there are four stages: stage 1, stage 2, stage 3 and stage 4. There is a convolutional layer between each two stages to reduce the number of markers and increase the number of channels, and the convolutional kernel size is $3 \times 3$ with a step size of 2. Finally, as a result of the four stages of feature mapping, the feature maps are $112 \times 112$, $56 \times 56$, $28 \times 28$, and $14 \times 14$ in size, the resolutions are 1/4, 1/8, 1/16, 1/32 of the original image, and the four channel numbers are 96, 192, 384 and 768. Our DMTC network uses the CSA mechanism to broaden the attentional scope and achieve global self-attentiveness. Additionally, in the self-attentive branch, we introduce a parallel module for Locally Enhanced Positional Encoding.

A cross-shaped window provides self-attention for horizontal and vertical bars, which is a major part of the mechanism. The input features map $T \in R^{(H \times W) \times C}$ under the action of a multi-head self-attention mechanism that first performs the linear mapping operation on m heads, and then the feature map obtained from each head mapping performs local self-attention using the CSA mechanism. For self-attention performed on horizontal bars, divide $T$ evenly into horizontal bars that are all $Sw$ in width, and these horizontal bars are non-overlapping. We denote these equal-width horizontal bars as $[T^1, T^2, \ldots, T^M]$, and each horizontal bar has $Sw \times W$ tokens. We use $Sw$ to adjust to balance learning ability and computational complexity. The horizontal bar's self-attentive output is:

$$T = [T^1, T^2, \ldots, T^M] \tag{1}$$

$$E_m^i = Attn(T^i W_m^Q, T^i W_m^K, T^i W_m^V) \tag{2}$$

$$H - Attn_m(T) = [E_m^1, E_m^2, \ldots, E_m^M] \tag{3}$$

where $T^i \in R^{(Sw \times W) \times C}$, $M = H/Sw$, $i \in [1, M]$. $W_m^Q \in R^{(C \times d_m)}$, $W_m^K \in R^{(C \times d_m)}$, $W_m^V \in R^{(C \times d_m)}$ are denoted as the $q$, $k$, $v$ projection matrices of the $m$-th head, respectively, and $d_m = C/M$. Finally, joining the outputs of the two parallel groups, horizontal and vertical bars together complete our global self-attentiveness, noting it as $CSAttn(X)$, and with:

$$CSAttn(X) = Concat(head_1, \ldots, head_m)W \tag{4}$$

$$head_m = \{ \begin{array}{l} H - Attn_m(T), m = 1, \ldots, M/2, \\ V - Attn_m(T), m = M/2 + 1, \ldots, M \end{array} \tag{5}$$

Since transformers use multi-heads in the computation of attention, in order to keep the amount of computation, this paper divides the head into two, one for row attention and one for column attention. There are 2, 4, 8 and 16 attention heads in each of the four stages of our DMTC network, and the $Sw$ are 1, 2, 7, 7, respectively. In Figure 2, we show the steps for implementing the attention manipulation mechanism for stage 1.

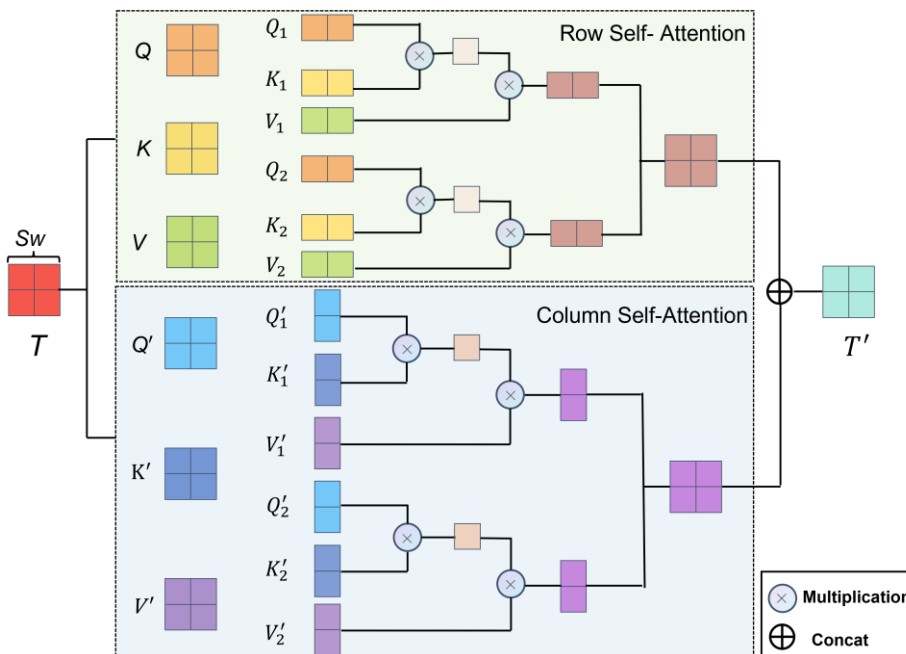

**Figure 2.** Architecture of the stage1. We use two parts to calculate the global attention of the feature graph T in stage 1; one part calculates the row attention and one part calculates the column attention. Finally, these two parts are combined to achieve parallel processing of the output feature map T'. Similarly, stage 2, stage 3 and stage 4 also output the corresponding attentional feature maps.

### 2.2. Dense Multiscale Feature Learning

Road damage has different shapes and scales. In comparison with large potholes, when potholes appear as multiscale targets at a distance, they are easily lost in the process of down-sampling because they contain less pixel information in themselves. For detection problems where the target size difference is very significant, we use the DMFL structure as a decoding module to solve the multi-scale fusion problem in road detection with a minimal computational effort. We apply the output feature maps obtained from the 4 stages of the CSAB model as the input of the decoding stage; their channel numbers are 96, 192, 384 and 768, respectively, and we denote these 4 output feature maps as $[C_2, C_3, C_4, C_5]$. For fusing the contextual information that feed the feature map into the DMFL structure, we add a $1 \times 1$ convolutional layer to $[C_2, C_3, C_4, C_5]$ to generate low-resolution feature maps $[P_2, P_3, P_4, P_5]$, where the feature map size remains the same and we reduce the channel count to 64, 128, 256, 512. Afterwards, we up-sample the feature mapping set $[P_2, P_3, P_4, P_5]$ so that they have the corresponding dimensions. To obtain the final fused feature map, we adopt an additive method to process the $[P_2, P_3, P_4, P_5]$ and their output after the up-sampling operation.

The shallow layer of the feature map has high resolution, and the deep layer has rich semantic information. We adopt the DMFL model to fuse the shallow layer and the deep layer, and increase the perceptual field of the shallow layer, so that the shallow layer can obtain more contextual information when performing multiscale target detection. As well as fusing multi-scale information, it also reduces the confounding effect caused by superimposing different scale feature maps, providing powerful semantic information and improving multiscale target detection accuracy.

### 2.3. Segmentation-Head

In order to generalize and learn the feature information from the previous stage and predict a better road damage segmentation map, we utilize a segmentation head module to generalize the previously learned features and implement the accurate image segmentation in this paper. Specifically, the module is a hierarchical convolutional structure. First, we downscale the fused feature maps using a convolutional layer with a $3 \times 3$ kernel size

and a 1 step size. This is followed by extraction of the feature map using a convolutional layer, with a convolutional kernel size of $3 \times 3$ and a step size of 1, in order to further generalize and learn the previous feature map, thus making the segmented image features more obvious and detectable. Lastly, we use a convolutional layer with a kernel size of $1 \times 1$ and a step size of 1 as a decision layer to reduce the dimensionality while correcting and reconstructing the feature map. To enhance the nonlinear relationship between the three convolutional layers, we introduce a rectified linear unit (ReLU) activation function between two convolutional layers. This also improves our DMTC network's nonlinear representation and feature fitting. The final layer of the neural network uses the Softmax function, which restricts the scores of all categories to be between [0,1] and the sum of the scores of all categories to be 1. This allows us to consider the final output as the probability of the category, so we can adjust the network by comparing the actual situation with the predicted situation.

### 2.4. Loss Function

Road damage presents a serious imbalance between the number of positives and negatives in one stage of analysis, and an unbalanced background before and after. In order to change this imbalance and improve the performance of the model for damage, we introduce a hybrid loss for flexible optimization, which combines the advantages of the Focal Loss [42] and the Dice Loss [43].

We use the Focal loss function to solve the problem of extreme imbalance between the number of positives and negatives, obtaining the Focal loss on the improvement of binary cross-entropy loss. It is a dynamically scaled cross-entropy loss with a dynamic scaling factor, which dynamically reduces the weights of easily distinguishable samples during training in order to focus the weights quickly on those that are difficult to distinguish. The formula is as follows:

$$L_{Focal} = -\alpha_t (1 - P_t)^\gamma \log(P_t) \tag{6}$$

where $P_t = \begin{cases} p, y = 1, \\ 1 - p, y \neq 1 \end{cases}$, $p \in [0,1]$, $\gamma \in [0,5]$ and weighting factor $\alpha \in [0,1]$, with weighting factor $\alpha$ when it is a positive sample and $1 - \alpha$ when it is a negative sample.

We can resolve the problem of before-after background imbalance by using the Dice Loss function, adopting the Dice coefficient to measure the similarity of points between two samples. The formula is as follows:

$$S = \frac{2|X \cap Y|}{|X| + |Y|} \tag{7}$$

where $X \cap Y$ denotes the intersection between $X$ and $Y$ samples, $|\ |$ denotes the number of elements, and the coefficient of the numerator is 2. In reality, the Dice loss refers to the negative value of the dice coefficient. The formula is as follows:

$$L_{DICE} = 1 - S = 1 - \frac{2|X \cap Y|}{|X| + |Y|} \tag{8}$$

The Dice coefficient is higher when loss is smaller, indicating that the two samples are more similar.

From the above loss function calculation, we can express the used hybrid loss as:

$$L = L_{FOCAL} + L_{DICE} \tag{9}$$

## 3. Experiments and Results

### 3.1. Datasets

The main types of road damage are cracking, block cracking, longitudinal cracking, transverse cracking, potholes and subsidence. Among them, cracks do not have much impact on people's normal life, the sinkage area is large and we can easy to find it. However,

potholes will further expand and eventually develop into sinkholes while affecting people's lives. Effective detection of road potholes is crucial, which is the main reason why we use them as test samples.

We use the Cracks-and-Potholes-in-Road-Images-Dataset [44] (CPRID) to represent the advantages of our DMTC network. The datasets contain a total of 2235 road images as well as label samples, each with a size of $1024 \times 640$. In our work, the image size of the dataset is $448 \times 448$ pixels, and the ratio between the training, validation, and test set samples is 3:1:1. The CPRID has a wide variety of information regarding potholes. Some of the pavement pothole images are shown in Figure 3.

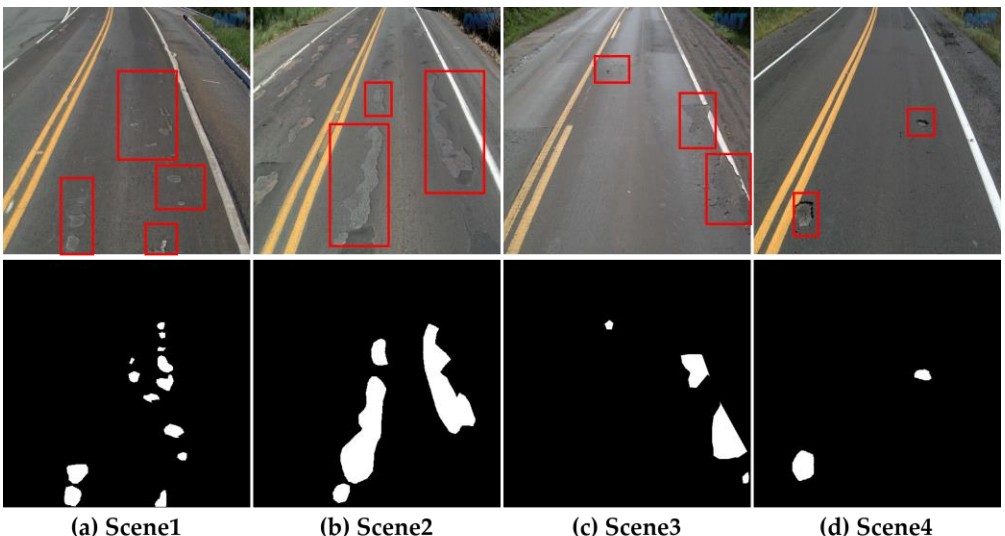

| **(a) Scene1** | **(b) Scene2** | **(c) Scene3** | **(d) Scene4** |

**Figure 3.** Visualization of the sample images from the CPRID. The first row is the source image and the second row is the label image. The red border crater in the first row of the source image corresponds to the white block pattern in the label image.

### 3.2. Experimental Details

### 3.2.1. Evaluation Metrics

We evaluate the proposed model using evaluation metrics: Precision, Recall, F1 score [45], Intersection Over Union_0 (IOU_0), Intersection Over Union_1 (IOU_1), Mean Intersection Over Union (mIOU), Overall Accuracy (OA), and Kappa coefficient. Specifically, the above evaluation metrics are calculated as follows:

$$Precision = \frac{TP}{TP + FP} \tag{10}$$

$$Recall = \frac{TP}{TP + FN} \tag{11}$$

$$F1 = \frac{2 \times TP}{2 \times TP + FP + FN} \tag{12}$$

$$IOU\_0 = \frac{TN}{TN + FN + FP} \tag{13}$$

$$IOU\_1 = \frac{TP}{TN + FN + FP} \tag{14}$$

$$mIOU = \frac{IOU\_0 + IOU\_1}{2} \tag{15}$$

$$OA = \frac{TP + TN}{TP + TN + FP + FN} \tag{16}$$

$$TMP = \frac{(TP+FP)(TP+FN)+(FP+TN)(FN+TN)}{(TP+TN+FP+FN)^2} \qquad (17)$$

$$Kappa = \frac{OA-TMP}{1-TMP} \qquad (18)$$

In the above formula, Precision represents the performance of evaluating error detection; higher precision means less frequent error detection. Recall represents the performance of evaluating missed detection, and higher Recall means less missed detection. The F1 score represents the efficiency of the algorithm. mIOU_0 represents the background IOU value; mIOU_1 represents the pitted IOU value.

### 3.2.2. Parameter Settings

We implement our DMTC using PyTorch, and perform all model training, validation, and testing experiments on an NVIDIA Tesla A100 GPU with 80 G of on-board memory. Before starting the model training, we complete the configuration of the parameters of the network model, conducting training in batches of four, with 500 rounds of training, and with a dynamic cycle learning rate by OneCycle method [46]. In addition, the potholes area occupies only a very small percentage of the entire image, and its background area has a more significant influence on the training of the model. Therefore, we preprocessed the original data by dividing the image into $448 \times 448$ blocks according to the ratio of $225 \times 225$, so that the similarity of the images is between [0,1]. This weakened the data noise and improved the stability of the model, thus improving the detection performance during training.

We use the same hyperparameters when training the CPRID with these network models, thus ensuring the fairness and validity of the algorithms. We do not change the original network parameters, but only change the network model. Therefore, it is also possible to compare the experimental results in a more intuitive manner.

### 3.3. Baselines

For the purpose of comparing the performance of our methods fairly, we reproduce 9 mainstream segmentation methods on the CPRID. The following is a brief overview of these models.

1. EfficientFCN [47]: An ImageNet pretrained network without any dilated convolutions forms the backbone of the system. Utilizing multi-scale features in the encoder to obtain high-resolution, semantically rich feature maps. To convert decoding tasks into novel codebook generation and codeword assembly tasks, encoders use their high-level and low-level functions.
2. IFNet [48]: A deeply supervised image fusion network. First, extracts features by using a full convolutional network with volume branching. The second step involves detecting changes using a deep supervised difference discriminative network.
3. UNet [10]: Getting the network by extending and modifying the full convolutional network. Two parts comprise the network: a contracting path for obtaining context information, and a symmetric expanding path for pinpointing the location.
4. SegNet [49]: A symmetric network consisting of encoder (left) and decoder (right). Encoder is a network model along the lines of Visual Geometry Group (VGG16), which mainly parses object information. The decoder converts the parsed information into the form of the final image.
5. FastFCN [50]: For the purpose of improving semantic segmentation, turning the extraction of high-resolution feature maps into a joint up-sampling problems by using a new joint up-sampling module JPU (Joint Pyramid Up-sampling).
6. PSPNet [51]: In this module, the core function is pyramid pooling, which aggregates context information on different areas in order to improve access to global information.
7. FCN16s [9]: The backbone network is Visual Geometry Group (VGG16), and the key step is to deconvolute (up-sampling bilinear interpolation can be done) the prediction

results of 1/32 graph into 1/16 graph. Predicts the results of the 1/16th graph pooling layer and adds them to the previous 1/32nd graph results. The final result is an enhanced version of the 1/16th graph prediction result, and then deconvoluting the predicted result to obtain the original image size to get the final outcome.

8. FCN32s [9]: In accordance with VGG16 (Visual Geometry Group) neural networks, removing the 3 fully connected layers firstly and then adding the 3 convolutional layers. To prevent overfitting, adding the dropout layers after each of the first 2 convolutional layers, and finally scaling up the results 32 times with transposed convolutional layers for restoring the original size of the output image.

9. DeepLabV3+ [52]: The model uses DeepLabv3 as the encoder module and a simple but effective decoder module as the decoder module. Through atrous convolution, the model can adjust the resolution of the encoded features, thus balancing accuracy and runtime.

### 3.4. Visual Performance

#### 3.4.1. Detection Results

Figure 4 represents the representative road damage segmentation results of our DMTC. In this paper, we provide six cases from the dataset CPRID, all including small potholes on the road, medium-sized potholes on the road, and large potholes on the road. In most cases, our DMTC is able to detect the actual pothole areas almost perfectly. Even if there are only minor road potholes in the scene images, as in columns 1–3 of Figure 4, our method is still able to accurately extract information about the road damage. For medium-sized potholes and large potholes, such as the scenes in columns 4–9 in Figure 4, our results are almost the same as GT as seen in the "Differ" images, which proves the powerful road pothole segmentation detection performance of the present network.

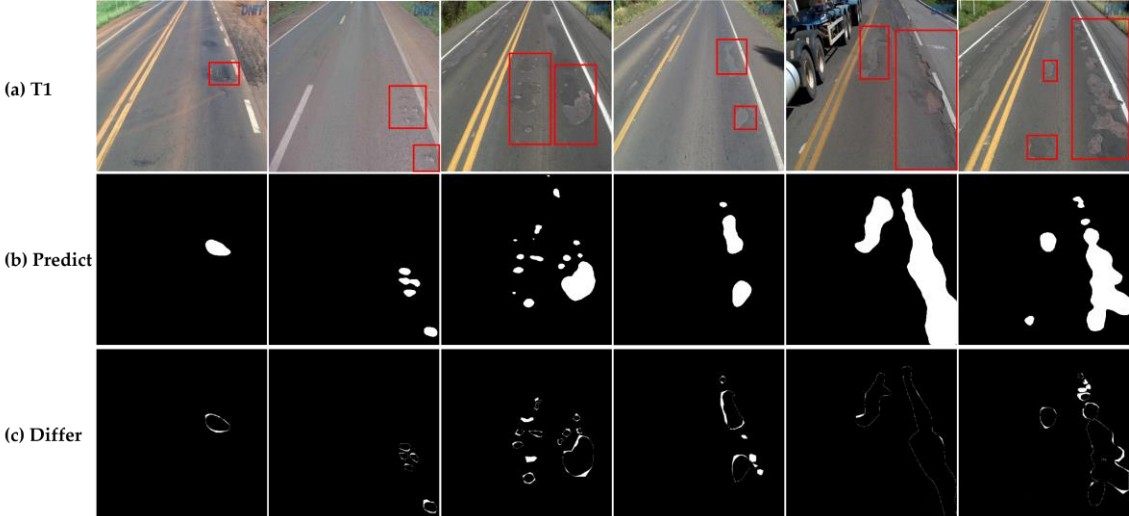

**Figure 4.** Visualization of segmentation test results on the CPRID. "Predict" represents the segmentation test results of our DMTC network and "Differ" represents the reduction between the predicted result and GT.

#### 3.4.2. Comparison with Baselines

To better demonstrate the segmentation performance of our DMTC in road damage task, we compared with other mainstream methods. In Figures 5–10, we visualize the results of the inspection experiments for some small potholes, medium-sized potholes, and large potholes. For these three types of images, our model basically reverts to a more regular boundary profile and a more compact interior when identifying single or multiple potholes. The other nine comparison network models are unable to accurately identify and detect road potholes. There are some models that are incapable of detecting potholes, such as DeepLabV3+, EfficientFCN, and FCN16s, and the remaining network models are

also prone to wrong results or missing detection cases. In our analysis of the entire set of images, our method shows a more intuitively superior performance.

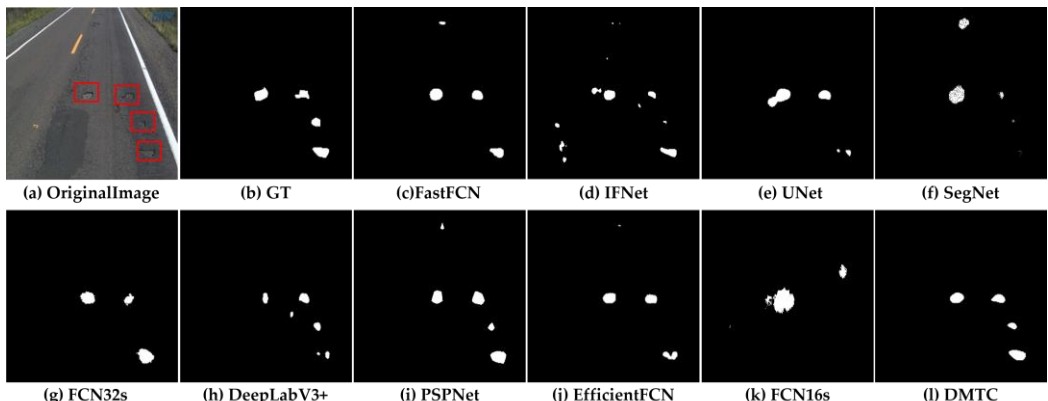

**Figure 5.** Visual comparison of the results of a series of segmentation methods for small potholes on CPRID. There is a red outline around the pits in the original image, a white outline around the pits in the label map and segmentation map, and black in the rest of the background.

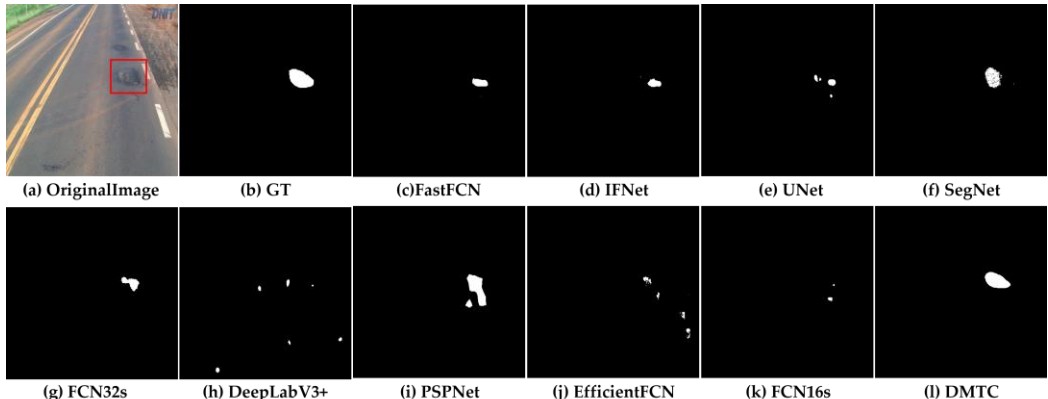

**Figure 6.** Visual comparison of the results of a series of segmentation methods for small potholes on CPRID. There is a red outline around the pits in the original image, a white outline around the pits in the label map and segmentation map, and black in the rest of the background.

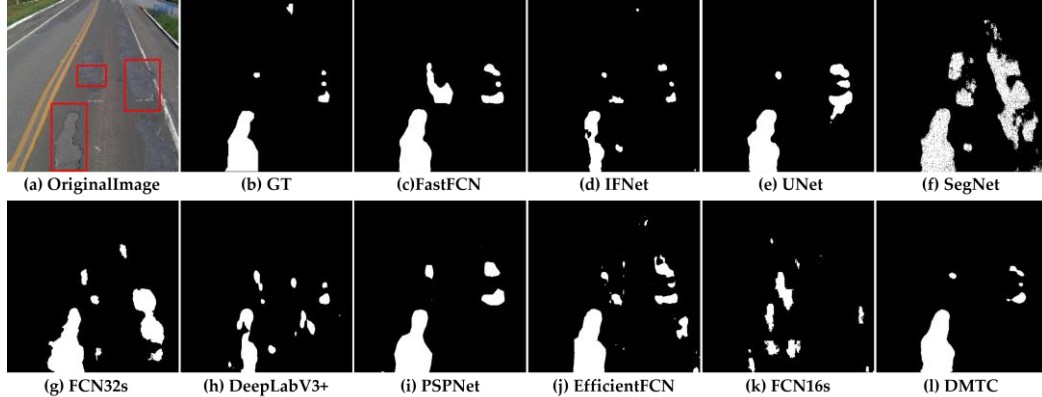

**Figure 7.** Visual comparison of the results of a series of segmentation methods for medium potholes on CPRID. There is a red outline around the pits in the original image, a white outline around the pits in the label map and segmentation map, and black in the rest of the background.

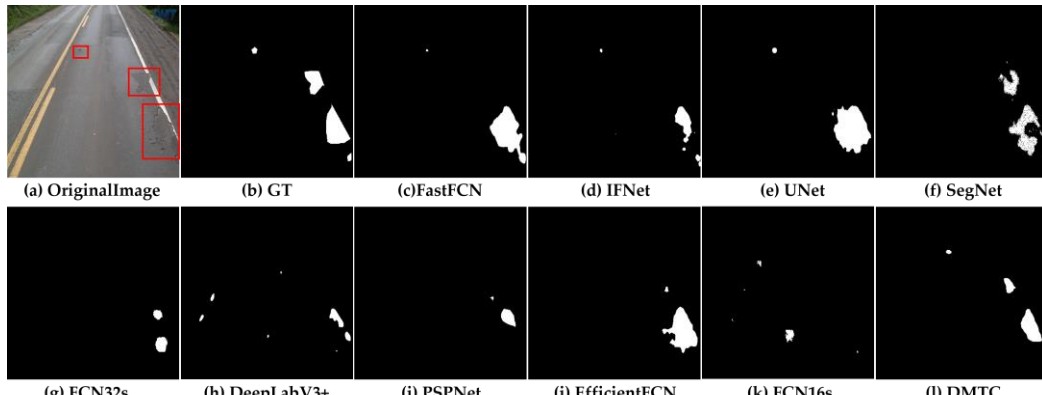

**Figure 8.** Visual comparison of the results of a series of segmentation methods for medium potholes on CPRID. There is a red outline around the pits in the original image, a white outline around the pits in the label map and segmentation map, and black in the rest of the background.

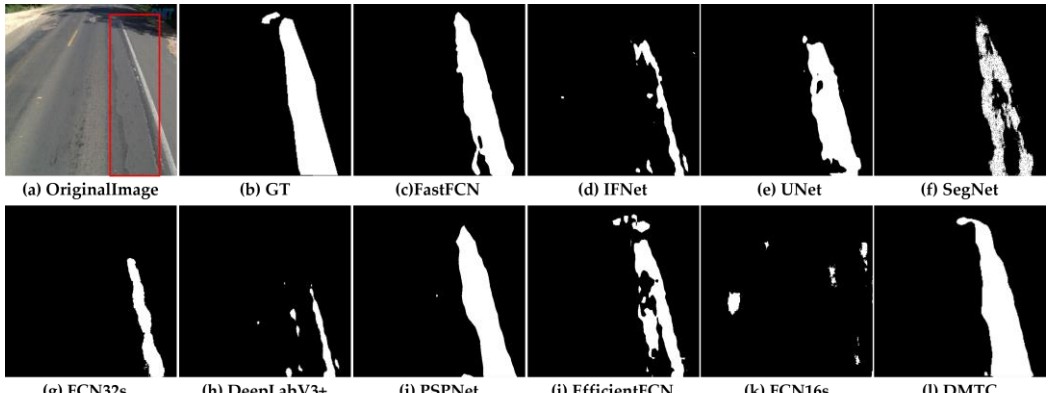

**Figure 9.** Visual comparison of the results of a series of segmentation methods for large potholes on CPRID. There is a red outline around the pits in the original image, a white outline around the pits in the label map and segmentation map, and black in the rest of the background.

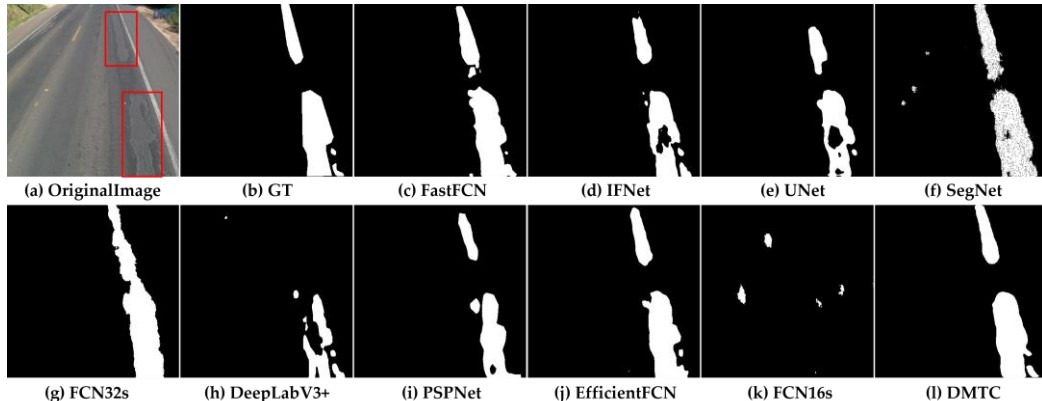

**Figure 10.** Visual comparison of the results of a series of segmentation methods for large potholes on CPRID. There is a red outline around the pits in the original image, a white outline around the pits in the label map and segmentation map, and black in the rest of the background.

*3.5. More Analysis*

3.5.1. Interpretable Analysis

In this section, we analyze the attention module by using the attention graph. Our DMTC is required to identify not only small, medium, and large potholes, but also other environmental factors. The size and shape of road potholes vary, making it challenging for

the network model to extract features pertaining to them, especially for small potholes. We use the CSA mechanism to perform the self-attentive calculation of pit shape size in parallel on horizontal and vertical bars, with each bar derived from splitting the input features into equal-width bars. Adjusting the width of the strips according to the depth of the network, and a wider stripe width can facilitate the connection between process elements. Thus, the presence of a continuous connection between the pitted features extracted at each layer of the network contributes to the network's model of attention. It may also increase the confidence that the network has learned to focus on the relevant features of the finely pitted images and to predict reasonable features. In Figure 11, we illustrate how our approach visualizes attention mapping at different stages through several examples of potholes. The blue color indicates that attention is low, whereas the red color indicates that attention is high.

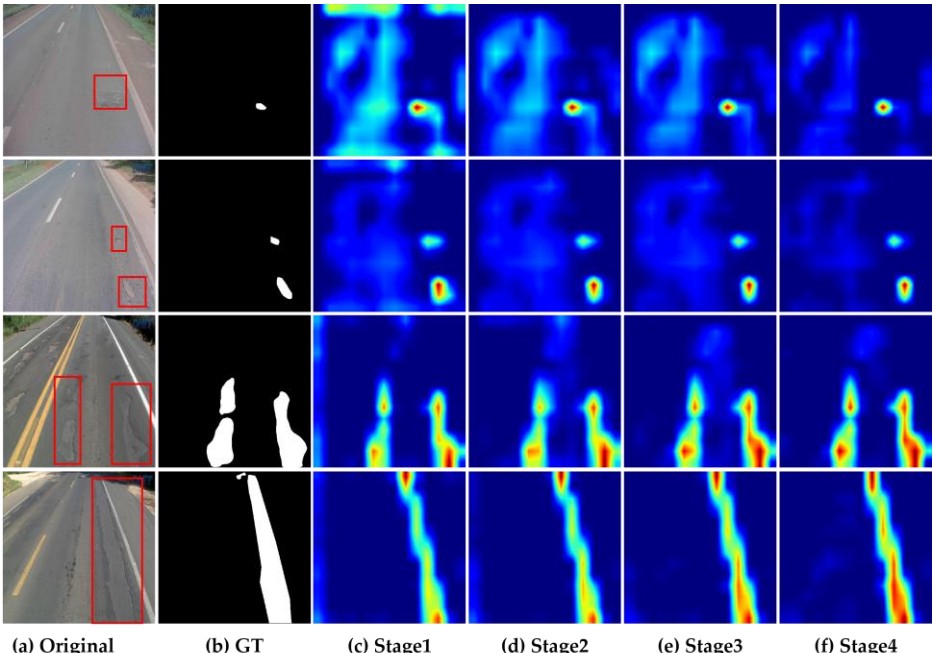

**(a) Original**    **(b) GT**    **(c) Stage1**    **(d) Stage2**    **(e) Stage3**    **(f) Stage4**

**Figure 11.** Attention map visualization of multiscale potholes in the CPRID. Blue indicates a lower attention value, and red indicates a higher attention value. There is a red outline around the pits in the original image.

As part of our analysis of CPRID, we create attention maps, and extract four images from the dataset for display. Our attention mechanism is a cross-shaped window self-attention channel. From Figure 11, we can observe that in the first and second scenes of CPRID, the small pit feature is not very obvious. The occupied area is relatively small. At this time, our attention mainly focuses on environmental information. In stage 1, the self-attentive channel also observes the small pit region, which was not extremely red in color. In spite of this, the CSA channel gradually becomes more focused on the small pit region as the network layer becomes deeper. At the fourth stage, our CSA channel peaks in the depressed area, with the contents of the depressed area showing up as dark red and the contents of the background area showing up as dark blue. Similarly, in the third and fourth scenes of CPRID, the attention module also reaches maximum attention in medium and large pothole areas.

### 3.5.2. Generalization Analysis

As the task of road damage detection is essentially a classification problem, and road construction is an indispensable part of our daily life, the damage detection is of vital significance for road maintenance and management. In order to prove that our DMTC has good generalization ability, we conducted test experiments on the crack data set in the

public road data set CPRID. Figure 12 shows that our method can also segment different sizes of cracks and obtain better segmentation results for road cracks.

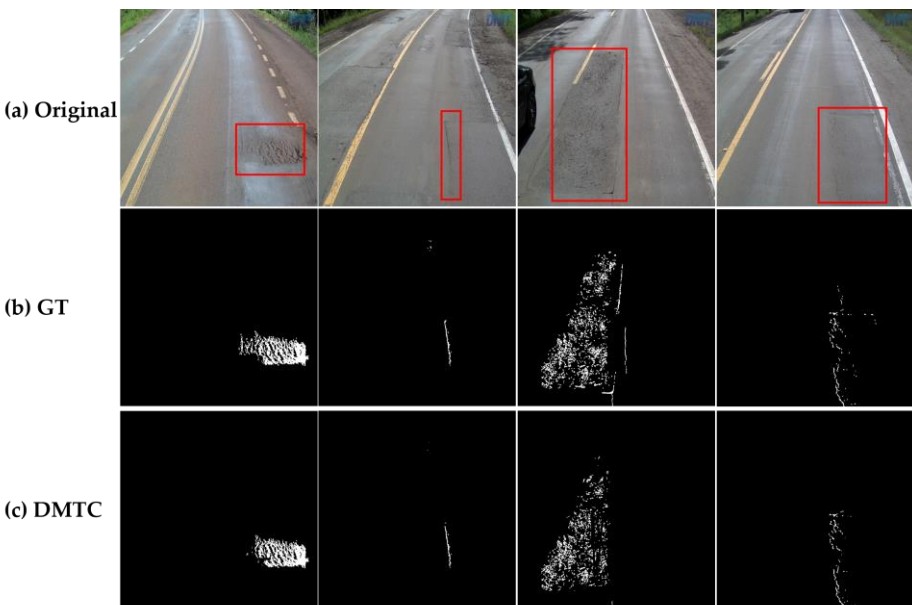

**Figure 12.** Visualization of partial crack segmentation results in the CPRID dataset. It includes the segmentation results of small, medium and large cracks. There is a red outline around the pits in the original image.

## 4. Quantitative Analysis

Table 1 shows the quantitative comparison between our DMTC and the most dominant algorithms. As can be seen, our network leads in almost all metrics for the multi-scale road damage image segmentation task. More specifically, we can see that our model outperforms the other models across all five aspects: *F1* score, *Precision*, *Recall*, *IOU_0*, and *Kappa*, achieving 79.39%, 81.37%, 77.51%, 65.83%, and 79.31%, respectively. In comparison with FastFCN that ranks second in comprehensive performance, our DMTC was 4.97%, 1.11%, 8.12%, 6.57% and 5% percent higher in these five metrics, respectively. In addition, although the values of *mIOU*, *IOU_1* and *OA* are close to the other nine comparison algorithms, the proposed model is still in an advantageous position in these three indicators. Overall, our method is the most effective.

**Table 1.** The quantitative results of proposed method on CPRID.

| Method | Precision | Recall | F1 Score | mIOU | IOU_0 | IOU_1 | OA | Kappa |
|---|---|---|---|---|---|---|---|---|
| EfficientFCN | 79.56 | 61.72 | 69.51 | 99.77 | 53.27 | 76.52 | 99.77 | 69.4 |
| IFNet | 64.17 | 40.56 | 49.7 | 99.65 | 33.07 | 66.36 | 99.65 | 49.54 |
| UNet | 76.52 | 59.4 | 66.89 | 99.75 | 50.25 | 75.00 | 99.75 | 66.76 |
| SegNet | 46.73 | 50.06 | 48.34 | 99.54 | 31.87 | 65.71 | 99.54 | 48.11 |
| FastFCN | 80.26 | 69.39 | 74.42 | 99.80 | 59.26 | 79.53 | 99.80 | 74.31 |
| PSPNet | 73.05 | 48.65 | 58.4 | 99.7 | 41.25 | 70.48 | 99.7 | 58.26 |
| FCN16s | 7.68 | 7.94 | 7.85 | 99.21 | 4.09 | 51.65 | 99.21 | 7.45 |
| FCN32s | 61.05 | 49.87 | 54.9 | 99.65 | 37.83 | 68.74 | 99.65 | 54.72 |
| DeepLabV3+ | 33.82 | 34.34 | 34.08 | 99.43 | 20.54 | 59.98 | 99.43 | 33.79 |
| **DMTC** | **81.37** | **77.51** | **79.39** | **99.83** | **65.83** | **82.83** | **99.83** | **79.31** |

Note that all metrics in the table are in percentages. The higher the value of these metrics, the better the performance. We highlight the best two results in red and green, respectively.

Moreover, we randomly selected some images from the 447 test set data for quantitative evaluation and comparison, and in Figure 13, we compare our model with the other nine comparison models in terms of the four metrics. Figure 11 illustrates that our

DMTC outperforms other method models in terms of *Recall*, *F1* score, and *mIOU* metrics. The accuracy of our method is lower than that of the UNet model, but still has significant advantages over the other remaining models.

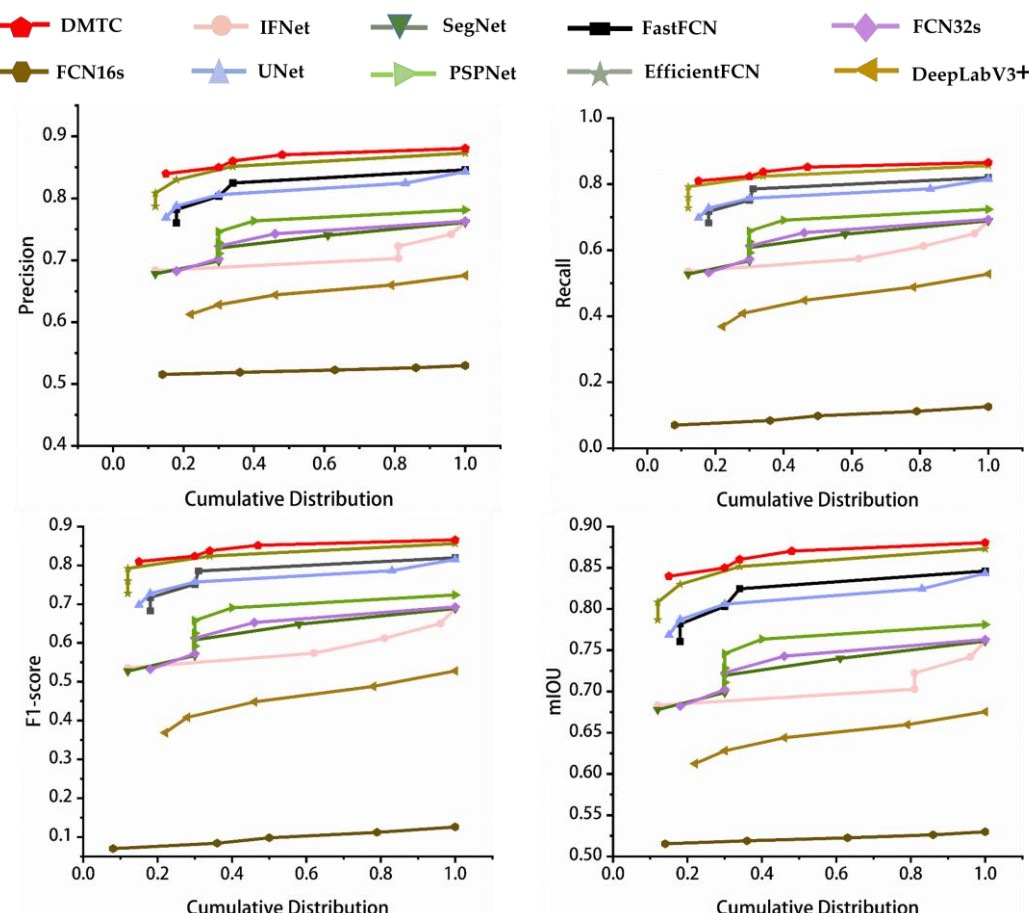

**Figure 13.** Quantitative comparison of four metrics, including precision, Recall, F1-score and mIOU, on the 5 random data images. Points (x, y) on the curve indicate that there are 100% * x% of images with metric values that do not exceed y.

## 5. Conclusions

We have proposed a DMTC network, which is a novel approach to road damage detection segmentation based on a cross-shaped attention mechanism and multi-scale feature fusion module. The CSAB module utilizes the CSA mechanism to expand the attention range, realize global attention, and resolve the problem of weak local feature extraction. The DMFL module fuses the extracted multi-scale features, which effectively solves the problem that multiscale targets are difficult to identify. Finally, using the multi-layer convolutional segmentation head, enables the network to summarize and learn the feature map. To show the superiority of our DMTC, we compared it with 9 mainstream semantic segmentation methods. Through the comparison experiment based on CPRID, we can find that the standard 8 index values obtained by our DMTC are higher than those obtained by other nine mainstream networks. From this, we have reached the following conclusions. Our DMTC has enhanced the ability of the network model to represent features through the cross-shaped attention mechanism. Meanwhile, introducing the dense multiscale feature learning module to fuse the multi-scale feature map by simple connection, greatly enhances the performance of the multi-scale pothole identification. In addition, we use the multi-layer convolutional splitting head to effectively summarize and learn the feature map, which has reduced the number of parameters and has improved network performance. At present, our model only performs segmentation experiments on potholes

of the road surface and achieves good experimental results. We intend to apply our network to future crack detection tasks, thus providing a contribution to road construction research.

**Author Contributions:** Conceptualization, C.X., Q.Z., L.M. and X.Z.; methodology, Q.Z.; software, Z.Y.; validation, L.M., S.S. and D.L.; formal analysis, X.Z.; investigation, L.M.; data curation, S.S.; writing—original draft preparation, Q.Z.; writing—review and editing, C.X.; visualization, L.M.; supervision, C.X., W.Y. and X.Z.; project administration, C.X.; funding acquisition, C.X. and W.Y. All authors have read and agreed to the published version of the manuscript.

**Funding:** This research was funded by National Natural Science Foundation of China (Nos. 41601443, 41771457); Scientific Research Foundation for Doctoral Program of Hubei University of Technology (BSQD2020056); Science and Technology Research Project of Education Department of Hubei Province (B2021351).

**Data Availability Statement:** Not applicable.

**Conflicts of Interest:** The authors declare no conflict of interest.

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
