# Peer review of "Dense Multiscale Feature Learning Transformer Embedding Cross-Shaped Attention for Road Damage Detection"

_electronics, doi:10.3390/electronics12040898_

Round 1

Reviewer 1 Report

The manuscript “ Dense Multiscale Feature Learning Transformer embedding Cross-Shaped Attention for Road Damage Detection” presents applied studies in the field of structural health monitoring. A dense multiscale feature learning transformer embedding cross-shaped attention for road damage detection network is proposed for automatic detection of road damage. The proposed model is evaluated on experimental results showing that the network could segment pavement pothole patterns more accurately.

The title is clear, concise and relevant. The abstract is representative of the work presented. Technical material is clearly presented in the text.

However, the research field on defect detection and identification is not new. The paper is one of many of the last years. The increase in accuracy compared to similar papers is rather moderate, since the other studies present also high accuracy.

On the other side, the presented defect detection does not solve the more important problem of recent defect prediction and prevention. In real problems, e.g. on common crossing contact surface, the problem is much more complicated due to many cracks initiation in the same area, but not all cracks grow to serious defects. Thus, it would be worthwhile (and recommended) to mention the studies with crack importance ranking and crack condition prediction (like evolution of rail contact fatigue on crossing , optical rail surface crack detection method based, prediction of rail contact fatigue on crossings etc that are also published in MDPI journals ).

The manuscript fulfills in general its aim and is recommended for publishing after minor revision. The additional remarks:

1)    The concept of deep learning usually assumes that the learning should be fully automatic without feature engineering. What could be the result metrics without it?

2)    Could the accurately detected damages be further classified depending on their severity/costs? Is the dataset labeled on the criterion?

Reviewer 2 Report

This paper presents a transformer neural network model base on encoding-decoding structure to detection segmentation of road pavement damage. The research uses the cross-shaped attention mechanism to expand the perceptual field of feature extraction and utilizes the dense multi-scale feature learning to integrate local information at different scales as well as multi-layer convolutional segmentation head to generalize the previous feature learning. The methodological novelty and high accuracy and effectiveness of the proposed model are the strengths of this study. On the other hand, not considering and testing the model on other pavement distress is one of the most prominent drawbacks of the article. The following modifications need to be revised:

1.     In the abstract section, the existing challenge and the research method should be clearly stated and the case study results of the paper should be presented briefly. Moreover, in this section, it's appropriate to provide the limitations and overall trend. This section should be presented in a more concise and effective way and needs to be revised.

2.     The introduction section (1) is not well organized, the main challenge of the study is not properly highlighted and extra content is listed. Totally, this section is a bit tedious and requires comprehensive and relevant information that is provided briefly.

3.     In the introduction section, briefly present the main aspects and focused points of the current review works. Summarize the differences between the current work and other existing review works that are also dedicated to the same research area. It is advisable to use the graphical overview or a table at the end of this section.

4.     In the introduction of your work, provide information about pavement distress and their types, and state your reason for choosing the pothole as the test sample.

5.     The assumptions considered in section 2 are not well supported. Please provide more information to support your work and assumptions.

6.     Please introduce the modules used in your model in full detail. It is also important to describe the proposed architecture in more detail and in an organized manner.

7.     What activation function have you used for the last layer of the neural network? This should be specified in your work and the reason should be mentioned.

8.     The information provided in Section 3 is insufficient and not well presented. This section could be rewritten more comprehensively, detailed and in an organized manner.

9.     State the reason for changing the size of the images used in the research. Have you used the technique of overlapping the resized images in order to avoid the loss of information?

10.  If pre-processing is done on the primary data, state them in the text of the article, and if not, provide the reason for not doing pre-processing.

11.  Provide the type and description of your model validation method. What solutions have been done to prevent data overfitting.

12.  It is recommended to display the information about the data set and model settings in table format for easier access of readers.

13.  Why have you considered the learning rate of the model to be constant? It is possible to improve the state of the model by choosing the dynamic learning rate.

14.  In order to improve the quality and reliability of your work, it is recommended to test your model on other datasets and with other types of pavement failures, and also include the summary of these reports in the text of the article (Section 4).

15.  The conclusion section (5) is not properly presented and should be more precise. To do this, it is suggested to mention the important and necessary points that related to the output of different parts of your results.

16.  The question is how useful these combination methods and models for segmentation of the road damage. In other words, the authors need to comment on the practical aspects of using such methods and support them

17.  It is suggested to provide more and better figures and tables in the article to provide a summary of the outputs of each part as well as the implementation method of different sections.

18.  It is better to reform the position of tables and graphs in the text of the article to be in a more optimal state.

19.  There are some typographical and formulation errors in the paper that need to be checked out and corrected. The English level should be improved.

20.  The format of all references is not the same. Moreover, the details of some references are not completely provided.
